

# Role of macrophages and their exosomes in orthopedic diseases

Riming Yuan and Jianjun Li

Shengjing Hospital, China Medical University, Shenyang, China

## ABSTRACT

Exosomes are vesicles with a lipid bilayer structure that carry various active substances, such as proteins, DNA, non-coding RNA, and nucleic acids; these participate in the immune response, tissue formation, and cell communication. Owing to their low immunogenicity, exosomes play a key role in regulating the skeletal immune environment. Macrophages are important immune cells that swallow various cellular and tissue fragments. M1-like and M2-like macrophages differentiate to play pro-inflammatory, anti-inflammatory, and repair roles following stimulation. In recent years, the increase in the population base and the aging of the population have led to a gradual rise in orthopedic diseases, placing a heavy burden on the social medical system and making it urgent to find effective solutions. Macrophages and their exosomes have been demonstrated to be closely associated with the pathogenesis and prognosis of orthopedic diseases. An in-depth understanding of their mechanisms of action and the interaction between them will be helpful for the future clinical treatment of orthopedic diseases. This review focuses on the mechanisms of action, diagnosis, and treatment of orthopedic diseases involving macrophages and their exosomes, including fracture healing, diabetic bone damage, osteosarcoma, and rheumatoid arthritis. In addition, we discuss the prospects and major challenges faced by macrophages and their exosomes in clinical practice.

## INTRODUCTION

Exosomes are widely present in blood, saliva, and urine, and almost all prokaryotic or eukaryotic cells secrete exosomes (*Barile & Vassalli, 2017*). Under an electron microscope, exosomes appear round in shape with a diameter ranging from 30 to 150 nm and an outer double lipid layer (*Shen et al., 2020*). The function of exosomes is primarily determined by the cell type. Exosomes mainly participate in intercellular information exchange, signal transduction, and changes in the metabolic state of tissues *in vivo* and play a role in cell migration and differentiation (*Phinney & Pittenger, 2017*). In addition, exosomes have been found to play crucial roles in the development and occurrence of cancer, bone diseases, and nervous system diseases. Recent research has suggested that exosomes carry lipids, DNA, and non-coding RNA that can modify the metabolic state or biological activity of tissues and cells, thereby preventing disease progression (*Kalluri & LeBleu,*

Corresponding author
Jianjun Li, 969560125@qq.com

2020). *Fu et al. (2019)* reported that exosomes secreted by gastric cancer cells promote tumor cell proliferation by activating the phosphoinositide 3-kinase/protein kinase B (PI3K/Akt) pathway. In addition, M2-like macrophage exosomes with a high expression of IL-10 inhibit alveolar bone absorption, inhibit osteoclast generation, and promote the osteogenic differentiation of MSCs in mice with periodontitis by activating the IL-10/IL-10R pathway, indicating that M2-like macrophage exosomes protect and repair bone tissue (*Chen et al., 2022*). Clinical studies have found that the levels of abnormally phosphorylated tau protein in exosomes extracted from the cerebrospinal fluid of patients with mild Alzheimer's disease (AD) are significantly lower than those in patients with severe AD. Exosomes containing phosphorylated tau protein have been shown to promote degenerative changes in neurons, providing an important basis for the early diagnosis of AD (*Saman et al., 2012*). In osteosarcoma, exosomes are also involved in tumor drug resistance because the tumor cells excrete the chemotherapy drugs through exosomes, resulting in a decrease in the accumulation of chemotherapeutic drugs in the tumor, leading to the development of tumor chemotherapy drug resistance (*Fu et al., 2023*). In addition, exosomes secreted by drug-resistant osteosarcoma cells contain permeable glycoprotein (P-gp) and drug resistance-associated coding RNA (MDR-1 mRNA), which are important factors for drug resistance in osteosarcoma cells (*Torreggiani et al., 2016*). Exosomes have been used as new markers and therapeutic targets for cancer diagnosis and prognosis in clinical practice; however, their widespread use in clinics is limited because of ethical concerns, technical difficulties, and efficiency issues.

In addition to their common components, such as microRNAs, IncRNA, mRNA, and lipids, exosomes also contain special components, such as ceramide and cholesterol, which play important roles in intercellular communication, tumor cell metastasis, and tissue metabolism (*Ariston Gabriel et al., 2020*). The most widely studied among them is miRNAs, which are used for the diagnosis and treatment of various diseases. Studies have shown that miR-378a is highly expressed in the exosomes of M2-like macrophages, promotes bone regeneration in the skull defect area, and generates a larger bone volume by activating the BMP2 signaling pathway (*Kang et al., 2020*). In addition, the M1-like macrophage exosome miR-21a-5p has been shown to promote the osteogenic differentiation of MSCs (*Liu et al., 2021a*). The key role of miRNAs in macrophage exosomes in bone regeneration provides important clues for the future immunotherapy of diseases such as fracture nonunion.

Macrophages are widely distributed throughout the body, including the alveoli, intestines, blood, and bone marrow, and play an indispensable role in the immune defense process, performing important functions such as host defense, tissue healing, and maintenance of tissue homeostasis (*Wang et al., 2019*). Macrophages can be divided into M1-like and M2-like macrophages, based on their different polarization states and functions, which have pro- and anti-inflammatory effects. With the development of technology, different macrophage subtypes can now be detected more accurately, refining macrophage typing. M2-like macrophages can also be subdivided into M2a, M2b, M2c, and M2d (*De Paoli, Staels & Chinetti-Gbaguidi, 2014*). M2a macrophages, also known as wound-healing macrophages, promote tissue healing mainly by secreting IGF and

fibronectin (*Wynn & Vannella, 2016*). M2b macrophages, also known as regulatory macrophages, mainly secrete a large amount of the anti-inflammatory factor IL-10 to reduce inflammation and thus reduce body damage (*Arora et al., 2018*). M2c macrophages, also known as acquired inactivated macrophages, exert pro-fibrotic activity by releasing TGF-β (*Rőszer, 2015*). Presently, the inflammatory reaction at the site of injury can be improved by regulating the polarization of macrophages, which is a potential method for treating diseases (*Yunna et al., 2020*). Among these, the stem cell exosome miR-451a targets MIF to regulate the polarization of macrophages towards the M2 type, thus promoting new bone formation and increasing bone volume in the bone-defect area (*Li et al., 2022b*). In osteoarthritis, M2a macrophages significantly improve degenerative changes at the lesion site (increase in the hyalinoid cartilage markers and decrease in the fibrochondral markers), reduce apoptosis and immune cell infiltration, and promote cartilage regeneration, which is crucial for maintaining synovial tissue homeostasis (*Liang et al., 2022*). In summary, it is important to recognize the powerful repair and regenerative promotion functions of M2-like macrophages, and it is anticipated that they will find clinical applications in the near future. Macrophages play an important role as immune cells *in vivo*. After reaching the injury site under the guidance of chemokines, macrophages are degraded, and the phagocytosed fragments and foreign substances regulate the immune response (*Varol, Mildner & Jung, 2015*). Macrophage function is disrupted when the numbers are abnormal or improperly activated, affecting tissue healing and exacerbating the local inflammatory response. For example, the release of many inflammatory factors and a decrease in the anti-inflammatory macrophages can lead to failure of tissue repair or fibrosis at the injury site (*Wynn & Vannella, 2016*). While secreting anti-inflammatory factors to alleviate reperfusion injury in the body, M2b macrophages also weaken the immune response, thus aggravating bacterial and fungal infections and promoting tumor cell metastasis (*Tsuchimoto et al., 2015*). In addition, studies have shown that macrophages are closely related to fracture healing, rheumatoid arthritis, osteosarcoma, and other diseases. Fracture healing requires reasonable coordination between the inflammatory cells and osteoblasts. In fracture healing, macrophages play an important role in the recruitment and differentiation of bone marrow mesenchymal stem cells (BMSCs). Bone morphogenetic proteins and vascular endothelial growth factor (VEGF) secreted by macrophages promote bone tissue healing at the injury sites (*Pajarinen et al., 2019*).

## SURVEY METHODOLOGY

We searched PubMed for relevant literature using the keywords (macrophages and exosomes) AND (fracture)/(macrophages and exosomes) AND (osteosarcoma)/ (macrophages and exosomes) AND (rheumatoid arthritis). Only research and review articles were included. Papers that were not articles or reviews were excluded.

### The role of macrophages and exosomes in fracture healing

With the aggravation of social aging, the risk of fractures among middle-aged and older individuals is increasing, particularly among menopausal women and older individuals

with osteoporosis. Simultaneously, fractures lead to a decline in the patient's quality of life and increase the social and economic burden (*Blank, 2019*). Clinicians remain perplexed by challenges associated with the healing of fractures. Despite advances in surgical technology and the rapid development of materials science, the problem of nonunion or delayed union of fractures persists because of various factors that affect the process of fracture healing, such as patient age, health status, fracture type, and treatment plan (*Giannoudis et al., 2018*). Macrophages play important roles in fracture healing (hematoma mechanization, callus formation, bone healing, and fracture remodeling). In the post-fracture inflammatory hematoma stage, inflammatory M1-like macrophages secrete various inflammatory factors, such as interleukin-1β and tumor necrosis factor-α. The immune system is activated to amplify the inflammatory response and induce osteoclast formation to engulf dead cells and tissue debris (*Batoon et al., 2017*). During the callus formation stage, anti-inflammatory M2-like macrophages begin to play a role, and the inflammatory response gradually weakens because a continuous inflammatory response is not conducive to fracture repair. During this period, M2-like macrophages recruit MSCs to the injured site by releasing OSM and bone morphogenetic protein-2 and inducing differentiation in the osteoblasts, thus generating new cartilage and bone tissue at the fracture site and causing the formation of cartilage callus (*Guihard et al., 2012*). During the advanced callus remodeling stage, the number of inflammatory M1-like macrophages decreases, while the M2-like macrophages play a major role in accelerating calcium salt deposition, matrix mineralization, and callus formation by secreting transforming growth factor-β, IL-10, and vascular endothelial growth factor (VEGF), thus accelerating tissue repair and healing (*Shin et al., 2021*).

When a fracture occurs, neutrophils reach the injured site and release various inflammatory factors and chemokines to recruit immune cells, including macrophages and mast cells (*Gu, Yang & Shi, 2017*). Macrophages are present at all stages of fracture healing. When macrophages are deficient in the early stages, fracture healing is affected, callus formation is slowed, and healing time is extended. During the early inflammatory stages, macrophages secrete OSM, which promotes intramembranous bone healing (*Guihard et al., 2015*). M1-like macrophages play an important role in hematoma inflammation during the early stages of fracture. Early in fracture, saporin conjugated Mac-1 antibody (Mac1SAP) is injected *in vivo* to deplete M1 macrophage populations, the expression levels of cytokines iL-1α, iL-6, and TNF-α in the body were reduced, and the imaging results showed that the fracture healing was not promising. This may be closely related to the changes in the cytokine profiles of the M1-like macrophages (*Hozain & Cottrell, 2020*). In addition, cytokine IL-6 secreted by M1-like macrophages can recruit MSCs to the fracture site in the early stages of fracture. After IL-6 was knocked out in the mouse femur fracture model, the cortical bone density and crystallinity at the fracture site were reduced, the amount of cartilage in the callus increased, and angiogenic disorders were also found (*Yang et al., 2007*). *Schlundt et al. (2018)* reported that the implantation of a collagen scaffold attached to IL-4 and IL-13 to induce the M2-like polarization of macrophages in a mouse fracture model improved bone regeneration, and the injured site showed a higher bone volume 21 days after healing, indicating the increase in M2-like macrophages

promotes bone formation. After the transgenic knockout of macrophages in 3-week-old mice, bone mineral density and trabecular bone density in the tibial fracture site of the mice with macrophage defects were reduced by 25% and 70%, respectively, compared to the control group. The number of BMSCs, cortical bone thickness, and bone deposition were also reduced, which is not conducive to fracture healing. This suggests that macrophages are crucial for maintaining the microenvironment at the fracture site and for promoting fracture healing during the early stages of fracture (*Vi et al., 2015*). Macrophages regulate the differentiation ratio of osteoclasts and osteoblasts by secreting cytokines such as bone morphogenetic protein 2/4 and TGF-β 1, which enhance bone formation (*Champagne et al., 2002*). A recent study focused on the role of macrophages in promoting the osteogenic differentiation of BMSCs. *Fernandes et al. (2013)* co-cultured a macrophage-conditioned medium with MSCs and found that the osteogenic differentiation of the MSCs was enhanced. They speculated that this was associated with the effects of macrophage-derived OSM (*Fernandes et al., 2013*). In addition, *Gong et al. (2016)* co-cultured M1-like and M2-like macrophages with MSCs in an indirect transwell to observe the osteogenic effect and found that the M2-like macrophage group had more bone-mineralized nodules and higher expression of alkaline phosphatase and osteogenic markers. This indicated that the osteogenic differentiation effect in this group was more significant. The content of pro-inflammatory factors in the conditioned medium of M1-like macrophages was higher, while the conditioned medium of M2-like macrophages mainly contained TGF-β, VEGF, and IGF-1. The authors suggested that an increase in growth factors in M2-like macrophages may contribute to osteogenic differentiation (*Gong et al., 2016*). A single cortical bone hole with a diameter of 0.8 mm was created on the surface of the tibial crest to construct a single cortical defect model of the mouse tibia. After macrophage scavenger receptor 1 (MSR1) was removed, delayed fracture healing was observed. Simultaneously, when the macrophages were co-cultured with BMSCs, the membrane receptor MSR1 promoted osteogenic differentiation of the BMSCs and maintained M2-like polarization by activating the PI3K/AKT signaling pathway (*Zhao et al., 2020*). Angiogenesis at fracture sites is an essential prerequisite for bone regeneration. The availability of oxygen and nutrients is more conducive to promoting fracture healing, and macrophages gathered in the granulation tissue at the fracture site are the main source of VEGF after early soft tissue injury; however, only some of these macrophages near the wound secrete VEGF, which is crucial for angiogenesis and fracture healing (*Wu et al., 2013*).

The role of exosomes in fracture healing has become a current research hotspot because exosomes, especially miRNAs, contain many substances that mediate intercellular information exchange and regulate cell proliferation and differentiation. miRNAs can be stably transcribed in BMSCs to promote osteogenic differentiation and accelerate fracture healing. When the low-expressed exosome miR-214-3p was co-cultured with MSCs, the mRNA expression levels of the osteogenic genes ALP, OPN, and BSP were increased, and the cell activity of the MSCs was improved. More bone trabeculae were observed in the femur and tibia, as well as increased bone mineral density and bone volume in the osteoporosis mouse model. Knocking down osteoclast genes to promote bone regeneration may be a potential treatment for delayed or nonunion fractures (*Zhu et al., 2018*). In some

studies, miR-5106 was found to be highly expressed in exosomes of the M2-like macrophages, and by directly targeting the salt-induced kinases 2 and 3 (SIK2 and SIK3) genes of the BMSCs, after co-culture with the BMSCs, the expression levels of osteocalcin, RUNX2, and alkaline phosphatase were increased, and mineral deposition was observed in the alizarin red staining experiment. In contrast, the experimental results for the M1-like macrophage exosome group were the opposite. In the mouse femur fracture model, 100 µg/ml of M2-like macrophage exosomes were injected locally at the fracture site on days 0, 4, and 7 after fracture. A larger callus volume, reduced fracture space, and significantly increased bone volume and bone mineral density were observed during imaging on day 7. Hematoxylin and eosin staining showed a decreased cartilage area and increased bone area at the fracture site, indicating that the M2-like macrophage exosome miR-5106 can accelerate fracture healing (*Xiong et al., 2020*).

With improvements in living standards, the incidence of diabetes has increased annually. Over time, patients with diabetes may start to experience more complications, including bone damage. Diabetes-associated bone damage has been attributed to the dysfunction of macrophages and BMSCs. In the delayed healing of diabetic fractures, the subtypes of macrophages in the microenvironment, mainly M1-like macrophages, are unbalanced. It is possible that the high glucose environment causes polarization of the macrophages towards M1-like macrophages, leading to a significantly prolonged inflammatory state in patients with diabetes. This imbalance has an inhibitory effect on bone regeneration (*Fahy et al., 2014*). In a bone injury model of diabetic mice, *Shen et al. (2021)* found an increase in the number of M1-like macrophages and a decrease in the number of M2-like macrophages. The protein expression levels of the osteogenic indices RUNX2 and OCN and the bone mineral density were lower than those of the healthy group on the 28th day after surgery, which may be due to the high glucose environment. The accumulation of glycosylated products and pro-inflammatory factors leads to a decline in the osteogenic ability of MSCs (*Shen et al., 2021*). Implantation of biological scaffolds containing bone morphogenetic protein-4 (BMP-4) at the site of the bone defects in diabetic rats regulates the polarization of macrophages to M2-like macrophages, thereby reducing the levels of inflammatory factors *in vivo*. New bone formation and increased bone volume were observed at the site of the bone defect 8 weeks after surgery. In addition, BMP-4 not only promotes the differentiation of MSCs into osteoblasts by activating the Smad signaling pathway but also induces the M2-like polarization of macrophages by enhancing the secretion of iL-10, which ultimately accelerates bone repair through the dual action of BMP-4 (*Sun et al., 2021*). Not all exosomal miRNAs promote osteogenic differentiation; among them, miR-144-5p plays an inhibitory role in fracture repair by targeting SMAD1. When the macrophage exosome miR144-5p collected in a high-glycemic environment was injected into the femoral fracture site of rats (200 µL was injected on days 1, 3, 5, and 7 after surgery), micro-CT and X-ray findings on day 21 after surgery showed the miR-144-5p group having smaller calluses, while the fracture space was larger, the bone volume/total volume value was significantly decreased, and protein levels of RUNX2, type I collagen, and OCN were also significantly decreased compared to the PBS group (*Zhang et al., 2021a*). Therefore, miR-144-5p, a macrophage exosome, can

impair fracture healing in a high-glucose environment. At the same time, we speculated that in a high-glucose environment, the uptake and action efficiency of exosomes that inhibit osteogenesis may be higher than that in a normal environment, and the presence of higher levels of inflammatory factors in the microenvironment, could also be one of the potential possibilities of diabetic bone damage. This is because, in the diabetic microenvironment, abnormal cell-to-cell communication occurs and abnormal amounts of exosomes are produced; these also provide potential therapeutic targets for bone damage caused by diabetes. In another report, *Wang et al. (2023)* found an increased proportion of M1-like macrophages and delayed local chondral formation and mineralization of fractures in a diabetic femoral fracture model, indicating a significant delay in fracture healing. After local injection of M2-like macrophage exosomes into the femoral fractures, the exosomes were found to induce M1-like macrophage differentiation into M2-like macrophages by activating the PI3K/AKT pathway. Decreased cartilage area and reduced fracture space were observed 21 days after surgery, indicating that M2-like macrophage exosomes can accelerate the healing of diabetic fractures (*Wang et al., 2023*). However, it is unclear which active substances in M2-like macrophages specifically promote osteogenic differentiation; this could be attributed to the action of miRNAs or proteins, highlighting the need for further study (Fig. 1).

## Progress in rheumatoid arthritis regulation by macrophages and exosomes

Rheumatoid arthritis (RA) is a chronic autoimmune disease with an unknown etiology. Anti-cyclic citrulline peptide antibodies (antibodies with cyclic citrulline polypeptide as an antigen have high sensitivity and specificity to RA and are one of the indicators for the early diagnosis of RA) and rheumatoid factors are frequently detected in the blood of middle-aged women (*Lin, Anzaghe & Schülke, 2020*). The main pathological features of RA are the invasion of immune cells, thickening of the synovium, formation of pannus (mainly composed of neovascularization, synovial cells, inflammatory cells, and cellulose, which mainly exists in the diseased joint cavity of RA), and destruction of articular cartilage and bone (*Huang et al., 2021*). In the late stage, it leads to joint deformity and rigidity, and in severe cases, it may cause disability, significantly impacting patients' lives (*Scott, Wolfe & Huizinga, 2010*). Simultaneously, it is accompanied by complications in other organs, such as atherosclerosis, anemia, pleural effusion, rheumatoid nodules, and Felty syndrome (a specific type of RA with an enlarged spleen and reduced amounts of white blood cells) (*Sayah & English, 2005*). Most patients can achieve remission if the disease is detected early, diagnosed, and treated on time (*Burmester & Pope, 2017*). Considering that RA cannot be cured, the primary treatment goal is to reduce inflammation and relieve pain. Currently, non-steroidal anti-inflammatory drugs such as aspirin and ibuprofen, which can reduce inflammation and relieve pain, and anti-rheumatic drugs such as methotrexate and hydroxychloroquine are mainly used as first-line treatments. Among these, methotrexate is the first choice (*Bullock et al., 2018*). Subsequently, due to the ongoing in-depth study of its pathogenesis and scientific development, better treatment methods and drugs could be developed in the near future.

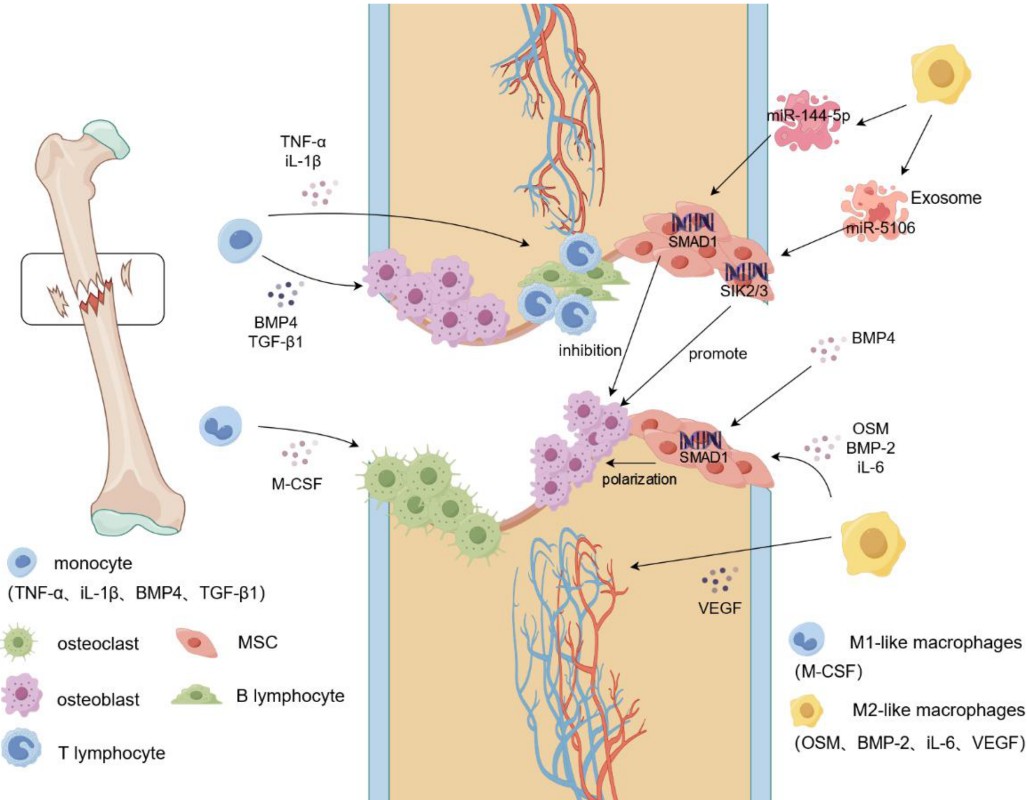

**Figure 1 Macrophage mobilization during fracture.** During a fracture, the integrity of the bone tissue is damaged, blood vessels rupture, and a hematoma forms locally in the fracture. Monocytes begin to secrete cytokines such as IL-1β and TNF-α to recruit immune cells, such as T lymphocytes and B lymphocytes, to the fracture site and produce an inflammatory response. Simultaneously, M-CSF induces additional monocytes to differentiate into osteoclasts and aggregate at the fracture site to clear the fracture fragments. Monocytes recruit osteoblasts to gather at the fracture site and promote osteogenic differentiation by secreting cytokines such as BMP4 and TGF-β1. In addition, M2-like macrophages secrete VEGF to induce vascular endothelial cells to form blood vessels, whereas OSM and BMP2 induce MSCs to reach the fracture site and generate new cartilage and bone tissue. Targeted activation of SIK2/3 by the M2-like macrophage exosome, miR-5106, promotes the differentiation of MSCs into osteoblasts, and miR-144-5p targets the activation of SMAD1 to inhibit osteogenic differentiation. TNF-α, tumor necrosis factors-α; iL-1β, interleukin-1beta; BMP-2/4, Bone morphogenetic protein-2/4; TGF-β, transforming growth factor-beta; M-CSF, Macrophage Colony Stimulating Factor; VEGF, vascular endothlial growth factor; iL-6, interleukin-6; OSM, oncostatin-M.

Normal synovial tissue includes synovial and sub-synovial linings, and different subtypes of macrophages originating from different sources are distributed in each synovial layer (*Smith et al., 2003*). Synovial macrophages are currently used as reliable biomarkers for the clinical assessment of RA severity. The sources of synovial macrophages include tissue-resident macrophages and blood-derived monocytes. Tissue-resident macrophages are divided into CD68+ and CD163+ macrophages, which are mainly distributed in the lining of the synovial membrane (*Tu et al., 2020*). CD68+ macrophages are the main subtype involved in the pathogenesis of RA, matrix degradation, and oxidative stress (*Harty et al., 2012*). CD163+ macrophages secrete VEGF and other cytokines and play a major role in angiogenesis, resulting in the formation of pannus and

other pathological tissues at the lesion site. Blood-derived monocytes expressing MRP8 and MRP14 accumulate in the synovium of patients with RA in large quantities, mainly in the synovium lining layer (*De Rycke et al., 2005*). Under normal conditions, the state of these cells remains relatively stable, and they are activated into M1-like and M2-like macrophages when lesions occur, which participate in inflammation and tissue metabolism *in vivo* (*Haringman et al., 2005*). Large amounts of macrophage infiltration have been reported in the synovium of patients with RA. Notably, different synovial macrophage subsets have different functions, among which infiltrating macrophages located in the sublayer of the synovium are the main cause of inflammation, leading to synovitis by secreting inflammatory cytokines such as IL-1 β, TNF-α, and IL-6, promoting the formation of osteoclasts, and aggravating bone destruction (*Li et al., 2020*). Many cell types, including neutrophils, B cells, macrophages, and mast cells, are involved in the pathogenesis of RA. Macrophages play important roles in the development of chronic inflammatory diseases. The degree of synovial macrophage infiltration positively correlates with the degree of joint damage. It is well established that alterations in the level of inflammatory cytokines in the microenvironment lead to polarization of macrophages into M1-like or M2-like phenotypes. Therefore, changes in the M1/M2 ratio indicate a shift in the disease condition, which can alleviate the disease or aggravate its progression. During the active phase of RA, the expression level of inflammatory factors increases, macrophages become polarized to the inflammatory M1-like phenotype, and the number of M1-like macrophages increases, thus, causing an increase in the ratio of M1/M2 macrophages. In addition, markers of M1-like macrophages were CD86, iNOS and iL-1β. The markers of M2-like macrophages were CD206, CD163 and Arg1, and the types of macrophages were determined by detecting these markers. The M1/M2 ratio positively correlates with the number of osteoclasts in the active phase. An increase in the M1/M2 ratio promotes the formation of osteoclasts in patients and causes diseased joint bone solvents and osteoporosis, thus aggravating the RA symptoms (*Fukui et al., 2017*).

The expression of macrophage-colony-stimulating factor (M-CSF) and receptor activator of nuclear factor κ-B ligand (RANKL) is increased in the microenvironment of patients with RA, as the precursor of osteoclasts; classical mononuclear macrophages increase the expression of the RANKL receptor on the cell surface during RA, which can promote the formation of additional osteoclasts (*Roszkowski & Ciechomska, 2021*). However, the role of macrophage-derived exosomes cannot be ignored. When plasmid DNA encoding IL-10 and betamethasone sodium phosphate (BSP) were embedded into M2-like macrophage exosomes and injected into the diseased joint cavity of a RA mouse model, the swelling and tetanus symptoms of the diseased joints were improved, and the levels of pro-inflammatory factors IL-12, IL-1β, and iNOS were significantly reduced (*Li et al., 2022a*). Moreover, compared to the model group, the average arthritis index of the RA mice injected with the M2-like macrophage-derived exosomes was lower, and their degree of dysfunction and inflammatory factors were reduced. These results indicated that macrophage-derived exosomes are good drug carriers that can promote RA treatment (Fig. 2).

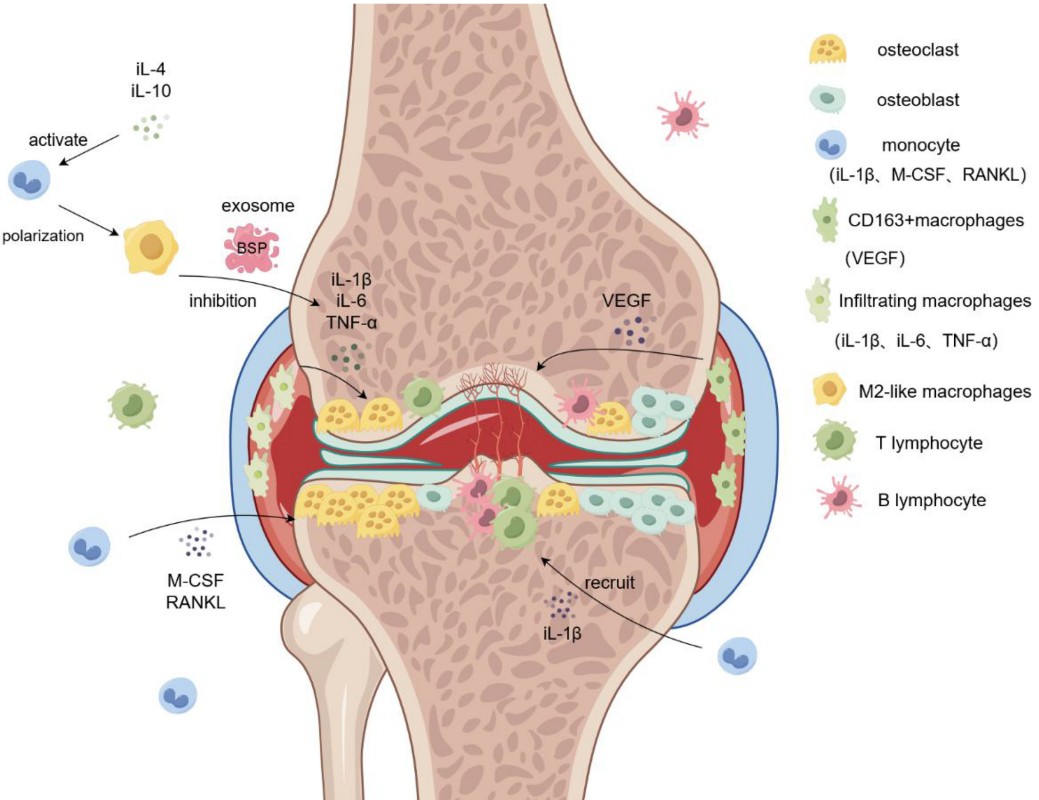

**Figure 2  Role of macrophages in rheumatoid arthritis.** Monocytes are induced by M-CSF and RANKL to form osteoclasts and aggregate in the diseased joints, leading to bone destruction and osteoporosis. Infiltrating macrophages in the synovium promote osteoclast formation and bone destruction by secreting IL-6 and TNF-α. M2-like macrophage exosomes containing BSP reduce inflammatory responses and inhibit bone destruction by reducing levels of the inflammatory cytokine IL-1β. Simultaneously, CD163+ macrophages secrete VEGF to promote angiogenesis and form pathological tissues, such as panni. iL-1β/4/6/10, interleukin-1beta/4/6/10; M-CSF, Macrophage Colony Stimulating Factor; RANKL, Receptor Activator of Nuclear Factor Kappa-B Ligand; VEGF, vascular endothlial growth factor; TNF-α, Tumor Necrosis Factor-α.               

## Macrophages and exosomes regulate the microenvironment of osteosarcoma

Osteosarcoma usually occurs in adolescents in the metaphysis of long tubular bones such as the distal femur and upper tibia. The disease recurs easily, and the survival time is short (*Yang, Zhang & Xu, 2019*). Osteosarcoma is affected by factors such as age, tumor size, health status, and treatment.

Osteosarcoma metastasis tends to occur in the lung, and a large number of M2-type TAMs have been found in the metastatic lung lesions. *Han et al. (2019)* demonstrated that TAM promoted the metastasis and local invasion of osteosarcoma cells by upregulating COX-2, MMP9, and p-STAT3 in cancer cells. At the tumor site, various immune cells gather and play different roles, among which NK cells inhibit tumor growth by recruiting dendritic cells to the tumor site. Osteosarcoma cells highly express CD54 and CD58, which are easily recognized and killed by NK cells (*Mariani et al., 1997*). Dendritic cells, as

antigen-presenting cells, are mainly responsible for presenting antigens to the T and CIK cells and stimulating their differentiation (*den Haan, Lehar & Bevan, 2000*). T cells secrete TNF-α and interferon and release perforin to kill osteosarcoma cells (*Paul & Lal, 2016*). TAM plays complex and diverse roles in osteosarcoma metastasis. Although it promotes tumor growth and protects cancer stem cells, it also inhibits tumor metastasis. This could be attributed to the role of TAM in the secretion of certain cytokines(COX2, VEGF, iL-10, TGF-β). However, the underlying mechanism remains unclear. Nevertheless, TAM may be a novel immunotherapeutic agent for osteosarcoma (*Zhao et al., 2021*). It releases epidermal growth factor (EGF) to allow cancer cells to escape from the primary site and grow in other organs. Simultaneously, cancer cells secrete M-CSF to act on the macrophages, causing the recruitment of more macrophages to the new metastatic sites, thus forming a positive feedback loop. However, this feedback mechanism can be blocked by inhibitors, resulting in reduced metastasis and cancer cells (*Joyce & Pollard, 2009*). In addition, TAM secretes immunosuppressive molecules such as TGF-β, Arg-1, and iL-10 to inhibit the host immune response, thereby promoting tumor escape and being an important link in cancer cell metastasis (*Terabe et al., 2003*). A recent study found that the intravenous injection of all-trans retinoic acid reduced the number of lymph nodes with lung metastasis of osteosarcoma and the expression of M2-type TAM in the lymph nodes, which may be related to the inhibition of MMP12 secretion by all-trans retinoic acid by M2-type TAM (*Zhou et al., 2017*). Reducing the recruitment of macrophages, removing them from tumor tissues, and promoting macrophage polarization can inhibit tumor tissue growth and metastasis. In the tumor microenvironment, IL-34 binds to the M-CSF receptor on the surface of macrophages, inducing the macrophages to transform into the M2-like phenotype and promoting additional M2-like macrophages to gather in the tumor tissue. At the same time, IL-34 promotes the proliferation and angiogenesis of vascular endothelial cells by activating the Akt and ERK1/2 signaling pathways and promotes the growth and metastasis of cancer cells through the aggregation and generation of new blood vessels by M2-like macrophages (*Ségaliny et al., 2015*). Matrix metalloproteinases released by TAM can promote osteosarcoma metastasis by degrading the matrix or enhancing epithelial-mesenchymal transition through activation of the NF-kB signaling pathway (*Lee et al., 2014*). Exosomes are important mediators of information between tumor cells and the extracellular environment because they contain several types of RNA and DNA. Researchers found that miR-29a was significantly reduced in osteosarcoma tissues as a tumor suppressor gene compared to normal tissues. lncRNA-LIFR-AS1 in macrophage-derived exosomes promotes osteosarcoma cell metastasis and reduces apoptosis by regulating the miR-29a/NIFA axis. By injecting macrophage exosomes into an osteosarcoma lung metastasis model, formation of osteosarcoma lung metastasis nodules increased, promoting the migration and proliferation of cancer cells. Moreover, NIFA is highly expressed in nodules (*Zhang et al., 2021b*). *Liu et al. (2019)* confirmed that miR-29a is a tumor suppressor. In *in vitro* experiments, the overexpression of miR-29a inhibited the migration and invasion of osteosarcoma cells by targeting the inhibition of CDC42 and significantly decreased the adhesion of cancer cells to plates (*Liu et al., 2019*). In another study, THP-1 cells were induced to become macrophages by adding PMA.

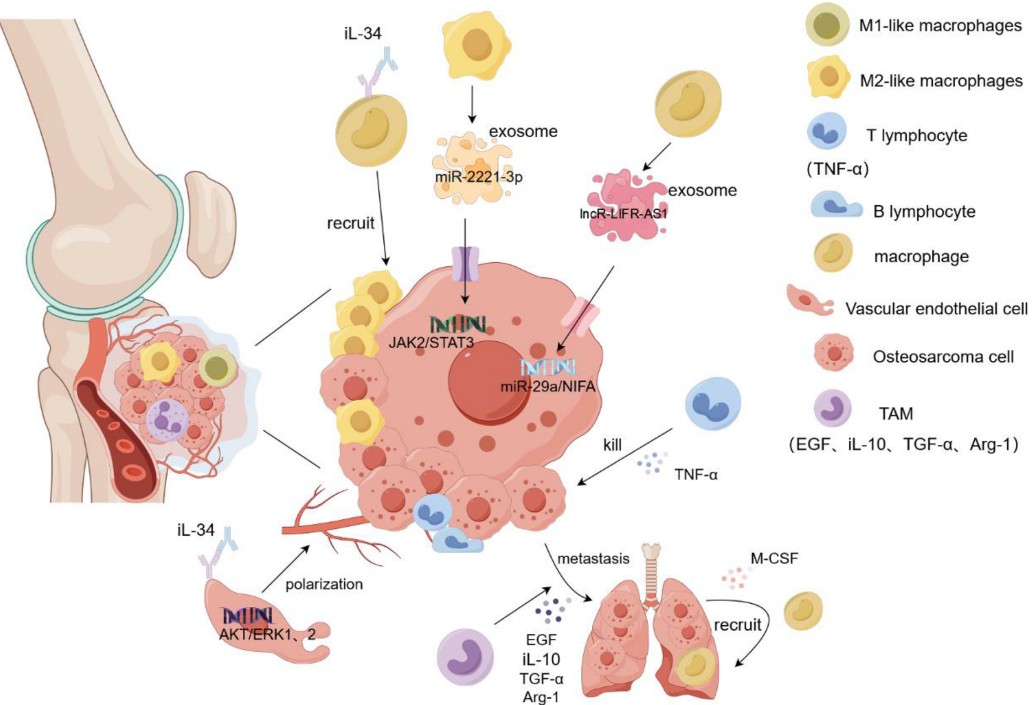

**Figure 3 Role of macrophages in osteosarcoma cell proliferation and metastasis.** Macrophages and their exosomes regulate tumor cell growth by regulating the tumor microenvironment. T cells secrete TNF-α to kill osteosarcoma cells. However, TAM secretes EGF and TGF-α to promote the metastasis of tumor cells to other organs, and tumor cells recruit additional macrophages to gather at the tumor site by secreting M-CSF. In addition, IL-34 recruits more macrophages to accumulate in tumor tissues by binding to the macrophage M-CSF receptor and promotes angiogenesis and tumor cell metastasis by targeting the activation of the ERK1/2 signaling pathway in vascular endothelial cells. The M2-like macrophage exosome miR-221-3p promotes the migration and invasion of tumor cells by activating the JAK2/STAT3 signaling pathway of tumor cells. Exosome LNCR-LIR-AS1 targets the miR-29a/NIFA axis of tumor cells to promote their proliferation and growth. iL-1β/10/34, interleukin-1β/10/34; TGF-α, transforming growth factor-α; M-CSF, Macrophage Colony Stimulating Factor.

Subsequently, macrophages were co-cultured with 143B and Saos2 osteosarcoma cells, and induced with iL-4 and iL-13 for 72h, then M2-TAM was screened. M2-polarized TAM-derived exosome miR-221-3p significantly enhanced the malignant behavior of osteosarcoma cells by activating the JAK2/STAT3 signaling pathway. *In vitro* experiments showed that miR-221-3p promoted the migration, invasion, and proliferation of cancer cells and hindered their apoptosis. Subcutaneous injection of TAM exosomes overexpressing miR-221-3p into an osteosarcoma mouse model showed that, compared to the blank control group, the tumor volume and weight were significantly increased in the overexpressed group, and the JAK2/STAT3 phosphorylation level was higher (*Liu et al., 2021b*). In summary, TAM and its exosomes play an important role in the metastasis and prognosis of osteosarcoma. However, many mechanisms that impact treatment and follow-up assessment remain unclear and warrant further clinical experimental studies (Fig. 3).

## CONCLUSIONS AND PERSPECTIVES

Macrophages are antigen-presenting cells that retain their plasticity and play key roles in inflammation and several diseases. As an important medium of communication between cells and the external environment, the contents of macrophage-derived exosomes are closely related to the occurrence and development of orthopedic diseases. Immunotherapy using macrophages has demonstrated advantages in orthopedic diseases. In particular, macrophages regulate the bone tissue microenvironment to promote bone tissue regeneration and repair by secreting various cytokines (VEGF, TNF-α, and IL-10) after polarization, and miRNA in exosomes inhibits metastasis and proliferation of osteosarcoma cells by regulating signaling pathways. However, this is not entirely without risk, and treating osteosarcoma with immunotherapy could result in the over activation of immune cells or immunosuppression, which can cause bacterial or fungal infections. However, bone immunity is still a relatively new research direction, and numerous unsolved scientific research questions persist, such as whether the uptake efficiency of exosomes by receptor cells is affected in a diabetic high-glucose environment, and how to ensure the complete targeted binding of macrophage exosomes to recipient cells in *in vivo* experiments. These questions will continue to be investigated in future research.

In addition, some chemical drugs and engineering materials that induce the polarization of macrophages or miRNA and drugs embedded in exosomes require clinical trials and human studies to ensure their safety and effectiveness because the polarization and mechanisms of macrophages and exosomes *in vivo* are extremely complex and cannot be reflected by simple *in vitro* or animal experiments.

At present, exosome studies for most orthopedic diseases mainly focus on miRNAs, especially in osteosarcoma metastasis and fracture healing, because exosome gene sequence detection has revealed that some miRNAs are specifically expressed or elevated, and disease can be cured through miRNA intervention. However, recent studies have found that the role of proteins in exosomes is equally important and can reflect the physiological or pathological state of the body's microenvironment, and they may participate in inflammation, cell information exchange, and tissue repair. For example, abnormal levels of phosphorylated tau protein in exosomes of the cerebrospinal fluid in patients with AD are crucial for the diagnosis and prognostic assessment of the disease; therefore, it is of great significance to focus on the abnormal changes of proteins in exosomes in both physiological and pathological conditions. However, the extraction and clinical application of exosomes are difficult. Currently, ultrafast centrifugation and magnetic bead methods are mainly used to extract exosomes; however, the exosomes obtained are not of high purity, and the investment cost is large. Therefore, it is important to develop efficient and inexpensive methods for exosome extraction. At present, some companies have begun to use exosomes in preclinical studies, among which Aegle Therapeutics plans to use BMSC-derived exosomes for the treatment of dystrophic epidermolytic bullosa. Studies on exosomes face challenges such as the unstable physicochemical properties of exosomes, unknown safety profiles, difficulties in obtaining ethical approval, short half-life, and immature drug carrier technology. Studies on

exosomes have been conducted to address these. Therefore, exosomes are not widely used in clinical treatment and diagnosis. We hope to overcome these limitations in the near future.

This study introduces the mechanism of action of macrophages and their exosomes in fracture healing, diabetic bone damage, rheumatoid arthritis, and osteosarcoma to explain how macrophages secrete different cytokines after polarization to regulate the bone microenvironment and promote bone tissue regeneration in fractures and diabetic bone damage. Exosomes inhibit osteosarcoma metastasis and reduce the inflammatory response in RA. At present, these diseases are mainly treated by surgery or drug intervention in clinical practice. However, there are some adverse reactions, such as fracture nonunion, immune system collapse caused by osteosarcoma chemotherapy, and infection caused by long-term hormone therapy. These adverse reactions necessitate the discovery of novel therapeutic strategies. Immunotherapy is a novel therapeutic strategy. Targeting and regulating the state of immune cells and the microenvironment promotes disease cure and has the advantages of a weak immune response and few adverse reactions. Although most of the experiments are basic research, they provide an important theoretical basis for treating these diseases in the future.

### Funding
This work was supported by the National Natural Science Foundation of China (No. 81971829). The funders had no role in study design, data collection and analysis, decision to publish, or preparation of the manuscript.

### Grant Disclosures
The following grant information was disclosed by the authors:
National Natural Science Foundation of China: 81971829.

### Competing Interests
The authors declare that they have no competing interests.

### Author Contributions
- Riming Yuan conceived and designed the experiments, performed the experiments, analyzed the data, prepared figures and/or tables, authored or reviewed drafts of the article, and approved the final draft.
- Jianjun Li conceived and designed the experiments, authored or reviewed drafts of the article, and approved the final draft.

### Data Availability
This is a literature review.

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
