# Peer review of "Role of macrophages and their exosomes in orthopedic diseases"

_PeerJ, doi:10.7717/peerj.17146_

## Round 0.1 · original submission · Major Revisions

Based on the considerable concerns of the reviewers, your manuscript is not yet acceptable for publication. However, if you are willing to provide a thorough response to the reviewers' comments, we would be happy to reconsider the manuscript. If you are willing to revise your manuscript, please include a letter detailing your response to each of the reviewer's comments.

**Language Note:** PeerJ staff have identified that the English language needs to be improved. When you prepare your next revision, please either (i) have a colleague who is proficient in English and familiar with the subject matter review your manuscript, or (ii) contact a professional editing service to review your manuscript. PeerJ can provide language editing services - you can contact us at copyediting@peerj.com for pricing (be sure to provide your manuscript number and title). – PeerJ Staff

Reviewer 1 ·

Basic reporting

I believe the topic of review is interesting and although highlighting a specific area of diseases it could provide valuable information to a broad audience of readers working in immunology and cancer biology and inflammation in both basic and translational research. However, the introduction does not provide sufficient information but it is rather vague and too general and needs to be extensively revised.

Experimental design

Sources are not adequately cited: authors need to provide more details about the studies cited.
Paragraphs are coherent, however paragraph related to “therapeutic approaches” do not provide any valuable info to readers as it does not involve the major topic of the review (exosomes) at all.

Validity of the findings

Conclusion does not contain speculative concepts or future perspectives.

Additional comments

• Please clarify “abstembranous”
• Abstracts is too vague.
• It would be important to expand more the introduction part regarding roles for exosomes in physiology and pathology, with particular focus in highlighting molecular mechanism of action for exosomes’ contents.
• Authors could give more information regarding the use of exosomes in clinical practice.
• It should be clarified which miRNAs are specifically involved in exosomes related to pancreatic cancer.
• What is the role of B-TrCP2?
• Macrophage introduction needs to be extensively revised and expanded. As it is, it appears too general and vague and referring to older literature of macrophage dichotomy M1/M2 which is considered outdated.
• Regarding fracture healing paragraph:
o statements on macrophages numbers and states in the stages of pathology results too vague and need to be expanded with more details.
o Authors should be more specific regarding the study (ref?) involving macrophage deficiency in early stages of fractures. Same applies to studies involving removing of macrophages from 3 week old mice.
o Please provide more info regarding the tibial defect model.
o It is not clear if VEGF in the model highlighted by the authors is only secreted by macrophages.
o Authors should provide more information on study in ref 21.
o What does miR-5106 target?
o Specify what alizarin red stain is.
o Authors are encouraged to better describe the methods and results from m1/m2 exosome injections in study in ref 22.
o The brief sentence of introduction to diabetes-related bone disease looks out of place. This topic should be better introduced.
o Please provide references for interplay between macrophages and BMSCs.
o Please clarify if miR-144-5p in ref 23 was injected and how. Moreover, regarding this study, authors should speculate if the high glucose environment is important for the efficiency of miR-144-5p.
• Regarding RA paragraph:
o Please clarify what citrulline peptides, pannus and fleet syndrome are.
o It would be important that authors provide more information regarding the different populations of macrophages distributed in the synovial layers.
o How is the ratio M1/M2 dictated by in RA?
o It is not clear if there is a feedback from osteoclasts back to macrophages that leads to increase proliferation of the latter.
o The part regarding activation of T cells could either be deleted or expanded with more details.
o Regarding ref 33: it is not clear M2 derived exosomes leading to less disfunction were injected purely or supplemented with IL10 and betametasone. Moreover, was this study able to identify what specifically in M2 exosomes is responsible for amelioration of disease?
• Regarding osteosarcoma paragraph:
o Please provide references for TAM correlation with metastasis and progression of osteosarcoma.
o It would be important to better explain the interplay of immune cells at the osteosarcoma tumor site.
o Please note that inducible nos enzyme is not secreted; to be secreted is its product nitric oxide.
o The part regarding TAM- related cytokines and IL34 is too general and vague. Please explain molecular mechanisms in detail.
o It would be important to give more details regarding the role of miR-21a and its involvement with osteosarcoma.
o Please reference the study involving miR-2213p.
o The summary of study in ref 40 is not clear; please provide more details.



• Paragraphs regarding “therapeutic methods” does not provide any substantial information and could be deleted. None of the sentences stated here refers to exosomes studies. Same applies to “conclusion” paragraph which does not contain authors ‘ speculations of future perspectives and could be deleted as well.
• Figures do not provide valuable and summarized information for the reader but they are rather not clear and vague. Authors in these should focus on exosomes contents and readouts on diseases: progression mechanisms and healing responses.

Reviewer 2 ·

Basic reporting

This topic is interesting and timely and is of broad interest. However, it has been an area that's been covered in several recent reviews (PMID: 37367263, 37303852). The role of macrophages in fracture healing has been extensively reviewed, so care should be taken in stressing novel viewpoints.

Experimental design

The survey methodology is not described, though a subject heading appears on line 68. There are sufficient references.

Validity of the findings

The article does not clearly lay out the objectives in the Introduction and is somewhat confusing because it spends a good amount of time discussing the roles of exosomes in pancreatic cancer, which seems off topic. It would benefit the article to have a clear and direct statement that is the central focus of the article. While the authors do describe tumor associated macrophages, it's unclear if this is related to the discussion of pancreatic cancer in the introduction.
The idea of macrophage polarization is over-simplified and the authors should discuss how macrophages contribute to repair.

Additional comments

1. There are concepts brought up in the introduction that seem off-topic for this review. For example, the authors discuss the role of mirco RNAs in diagnosing pancreatic cancer. As this article is focused on orthopedic diseases, one suggestion is to find a similar example that's more relevant to bone/fracture healing.
2. The term "abstembranous" is not commonly used and should be defined.
3. Macrophage polarization is more nuanced than M1 and M2 and the idea that M2 macrophages are anti-inflammatory is omitting the importance of reparative functions.
4. Perhaps change wording (line 228) from "Macrophages are plastic antigen presenting cells..." to "Macrophages are antigen presenting cells that retain plasticity..._

---

## Round 0.2 · Major Revisions

The original Academic Editor is not available so I have taken over handling this submission.

As you can see, both reviewers are not completely satisfied with your revision. Therefore, your manuscript requires additional work to address the remaining concerns of the reviewers.

Reviewer 1 ·

Basic reporting

Authors have much improved the manuscript and figures. However, I still have some points of concern.

1. In the figures please specify which cells are responsible for secretion of which factor.
2. It is not clear what are the evidences that Nitric Oxide has a direct role in recruiting immune cells.
3. It would be important that authors tried to avoid the macrophage nomenclature M1/M2a/M2b/M2c. It is nowadays pretty much surpassed and outdated as macrophages especially in in vivo settings can assume a variety of phenotypes that overlap one or more or those phenotypes mentioned. Authors could still reference this older literature though.
4. Importantly when talking about M1/M2 throughout the manuscript, please rephrase as “M1-like” and “M2-like”, especially when talking about cancer or disease rather than in vitro, and highlight their phenotype based on gene expression and markers during the pathology described.
5. Please describe how in “Hozain & Cottrell 2020” specifical M1 depletion is achieved.
6. Regarding M1/M2 in RA, please describe what cytokines and markers are used in this disease to define these macrophages.
7. In citing “Liu et al. 2021b” please explain how M2-polarized TAM are generated.

Experimental design

The content is very much in the scope of the journal.

Validity of the findings

I believe the review is "novel" in the sense that a summary of findings regarding macrophages specifically in this context is not common and could provide benefit for a broad audience. Conclusions have been improved and better discuss therapeutic implications.

Reviewer 2 ·

Basic reporting

Yes, this review is of broad interest and the revised manuscript provides a much better Introduction to the topic.

Experimental design

The sources are adequately cited and details are provided on the literature that was used.

Validity of the findings

The revised manuscript is much more refined and stays on topic.

Additional comments

The rewritten manuscript is much improved. It now provides a more inclusive description of the field and is more focused to the topic outlined in the Introduction.

---

## Round 0.3 · accepted · Accept

All concerns of the reviewers were adequately addressed and the revised manuscript is acceptable now.

Reviewer 1 ·

Basic reporting

manuscript is further improved

Experimental design

concerns have been addressed

Validity of the findings

no further comments